# Chemical modification of proteins by insertion of synthetic peptides using tandem protein trans-splicing

K. K. Khoo [1,3], I. Galleano[1,3], F. Gasparri [1], R. Wieneke [2], H. Harms[1], M. H. Poulsen[1], H. C. Chua [1], M. Wulf[1], R. Tampé [2] & S. A. Pless [1✉]

Manipulation of proteins by chemical modification is a powerful way to decipher their function. However, most ribosome-dependent and semi-synthetic methods have limitations in the number and type of modifications that can be introduced, especially in live cells. Here, we present an approach to incorporate single or multiple post-translational modifications or non-canonical amino acids into proteins expressed in eukaryotic cells. We insert synthetic peptides into GFP, Na$_V$1.5 and P2X2 receptors via tandem protein trans-splicing using two orthogonal split intein pairs and validate our approach by investigating protein function. We anticipate the approach will overcome some drawbacks of existing protein enigineering methods.

[1] Department of Drug Design and Pharmacology, University of Copenhagen, Jagtvej 160, 2100 Copenhagen, Denmark. [2] Institute of Biochemistry, Biocenter, Goethe University Frankfurt, Max-von-Laue Strasse 9, 60438 Frankfurt/Main, Germany. [3]These authors contributed equally: K. K. Khoo, I. Galleano. ✉email: stephan.pless@sund.ku.dk

Chemical or genetic engineering of proteins provides great potential to study protein function and pharmacology or to generate proteins with novel properties. However, despite recent technical achievements[1,2], the type of chemical modification that can be accomplished by genetic means (e.g., amber codon suppression) is limited to incorporation of non-canonical amino acids (ncAAs) due to the tolerance of the cell's translational machinery. Additionally, insertion of multiple chemical modifications by genetic code expansion remains a challenge, particularly in eukaryotic cells. Semi-synthetic approaches offer an alternative means to manipulate proteins post-translationally, but these modifications have typically been performed in vitro[3–8]. We thus sought to complement these approaches with a method that could incorporate synthetic peptides carrying multiple post-translational modifications (PTMs) or ncAAs into both cytosolic and membrane proteins in live eukaryotic cells.

Split intein pairs comprise complementary N- and C-terminal intein fragments (Int$^N$ and Int$^C$) that assemble with extraordinary specificity and affinity to form an active intein. This assembly results in a spontaneous, essentially traceless splicing reaction that covalently links the two flanking protein segments through native chemical ligation[9]. The critical requirement for splicing to occur is typically the presence of a Cys, Ser, or Thr side chain (depending on the split intein in question) in the +1 position of the extein (the sequence flanking the split intein), and multiple split inteins have recently been optimized for increased splicing efficiency[10–12]. The latter facilitates the simultaneous use of two orthogonal split inteins within the same peptide or protein, an approach termed tandem protein trans-splicing (tPTS). However, tPTS has largely been conducted in vitro or restricted to bacterial expression systems, cell lysates, nuclear extracts, or selection protocols[8,13–15]. Indeed, most live-cell applications of PTS utilize single split inteins for the purpose of N/C-terminal tagging[16–18] or manipulating protein assembly/expression[19,20].

Here, we employ tPTS using two orthogonal split-intein pairs to insert synthetic peptides into proteins between two splice sites (A and B). This approach permits the introduction of a virtually limitless array of modifications, including PTMs, PTM mimics, and ncAAs, into live eukaryotic cells and allows multiple modifications to be made simultaneously (Fig. 1). We validate our approach by using tPTS to modify GFP, intracellular regions of the Na$_V$1.5 channel, and the extracellular domain of the P2X2 receptor, allowing us to gain insight into the role of PTMs and PTM mimics in ion channel function and the importance of spatial positioning of charge in ligand sensitivity.

## Results

### Post-translational incorporation of synthetic peptides.
Our goal was to generate semi-synthetic proteins in live eukaryotic cells by post-translationally incorporating ncAAs or PTMs into a protein of interest. We achieved this by using two orthogonal split inteins (A and B) to insert a synthetic peptide carrying these modifications. We designed three fragments of the protein of interest (Fig. 1), corresponding to N and C-terminal fragments (N and C), and a shorter central fragment containing the desired modification (peptide X). Fragments N and C were heterologously expressed in the cell, while peptide X was generated synthetically and inserted into the cell via an appropriate technique (e.g., injection). To covalently assemble the three fragments, the highly efficient engineered derivative of the NpuDnaE split intein (termed CfaDnaE)[11] was employed as split intein A. The first 101 amino acids of its N-terminus (Int$^N$-A) were expressed as a fusion construct at the C-terminus of protein fragment N. The corresponding C-terminal part (Int$^C$-A), consisting of amino acids 102–137, were attached to the N-terminal end of peptide X.

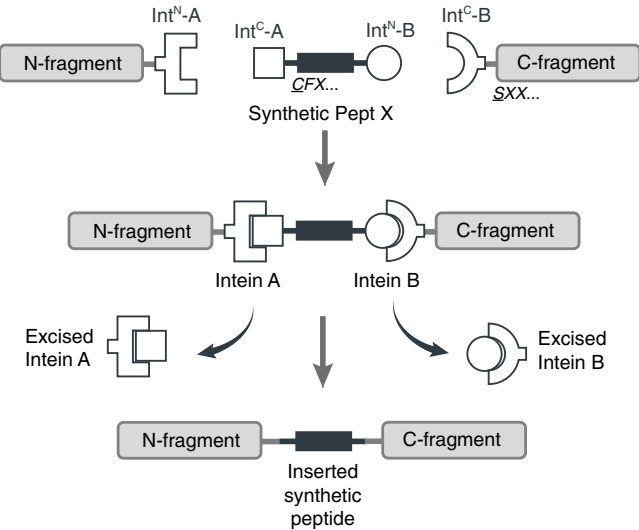

**Fig. 1 Schematic diagram of the tPTS strategy to incorporate recombinant and synthetic proteins.** The chosen split inteins used in this study, inteins A (CfaDnaE) and B (SspDnaB$^{M86}$), are indicated by square and round symbols, respectively. Flanking +1, +2, +3 extein residues for each intein are indicated in italics at their respective positions in the top panel. The +1 extein residue (underlined) is a critical requirement for splicing to occur. X denotes that the type of residue at that position is not critical for splicing, although they might affect the kinetics or splicing efficiency.

The optimized split-intein SspDnaB$^{M86}$ (ref. [10]) was chosen as split intein B because it can be split highly asymmetrically and has previously been shown to be orthogonal to the NpuDnaE split intein[21]. Its N-terminal part (Int$^N$-B), comprising only the first 11 amino acids, was added to the C-terminus of peptide X. The corresponding C-terminal part (Int$^C$-B), consisting of amino acids 12-154, was expressed as a fusion construct at the N-terminus of protein fragment C (Fig. 1).

### Replacing Na$_V$1.5 interdomain linkers with synthetic peptides.
To demonstrate the feasibility of our approach, we chose the well-characterized cardiac voltage-gated sodium channel isoform Na$_V$1.5, which is crucial for the initiation and propagation of the cardiac action potential[22]. This large 2016-amino acid protein comprises four homologous domains (DI-DIV) that are connected by intracellular linkers (Fig. 2a). Dysfunction of Na$_V$1.5 can arise from mutations, as well as dysregulated PTMs. For example, acetylation of K1479 and changes in phosphorylation of the linker between DIII–DIV have been shown to play a role in cardiac disease[23,24]. However, interrogation of the role of PTMs has been hampered by the inability to express a homogenous population of channels containing a defined number of PTMs in living cells. Thus, although phosphorylation at Y1495 is known to affect channel function[25] and phosphorylation can be prevented in a channel population by mutating Y1495 to phenylalanine, this population cannot be compared with one that is fully modified because the extent of phosphorylation cannot be controlled. Similarly, it is not known if there are synergistic effects with other PTMs in the vicinity, e.g., acetylation of K1479.

We tested the tPTS strategy by reconstituting full-length Na$_V$1.5 from three recombinantly co-expressed channel fragments. To this end, we designed three different gene constructs: (1) an N-terminal construct (N) comprising amino acids 1–1471 of Na$_v$1.5 (equivalent to DI-III) fused to the N-terminal part of CfaDnaE (Int$^N$-A); (2) a C-terminal construct (C) corresponding to the C-terminal part of SspDnaB$^{M86}$ (Int$^C$-B) linked to the C-

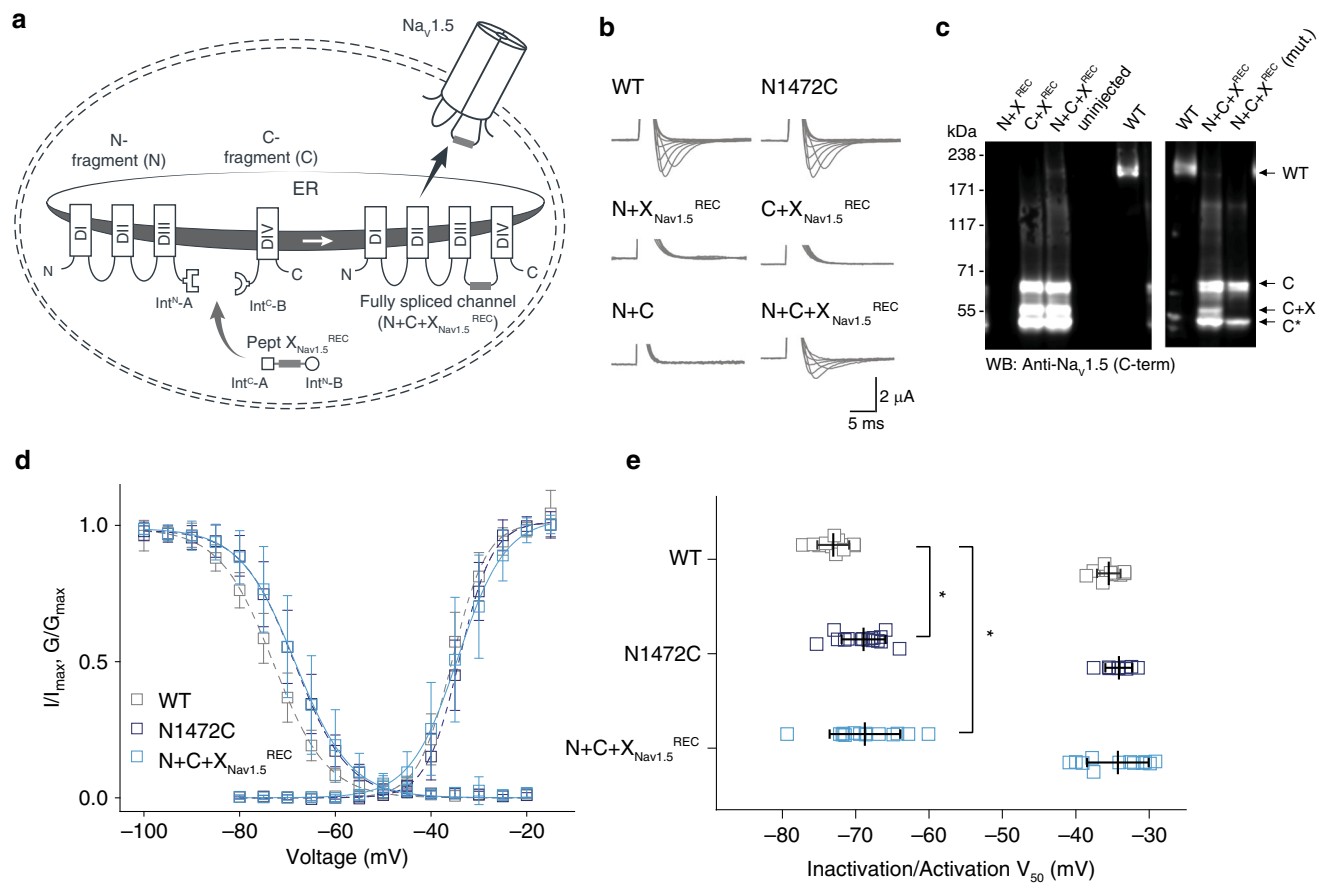

**Fig. 2 Insertion of recombinantly expressed peptides into the DIII–IV linker of NaV1.5. a** Schematic presentation of the strategy to reconstruct full-length $Na_V1.5$ from recombinantly expressed N-/C-terminal fragments (N and C) and a recombinantly expressed peptide corresponding to amino acids 1472 to 1502 of the $Na_V1.5$ DIII–DIV linker (X) in *Xenopus laevis* oocytes. Inteins A (*Cfa*DnaE) and B (*Ssp*DnaB[M86]) are indicated by square and round symbols, respectively. Note that we cannot exclude the possibility that splicing takes place at a different subcellular location than depicted here. **b** Representative sodium currents in response to sodium channel activation protocol (see methods; only voltage steps from −50 to +10 mV in 10 mV steps are displayed), demonstrating expression of functional $Na_V1.5$ only in the presence of all three components (N + C + X), along with WT and N1472C channels (the latter mutation was introduced to create an optimized splice site for intein A. **c** Immunoblots verifying the presence of fully spliced $Na_V1.5$ only when all three components (N + C + X) were co-expressed (using antibody against $Na_V1.5$ C-terminus). $Na_V1.5$ band was not detected when one component was missing (left blot) or when non-splicing mutation (N + C + X mut.) was introduced to prevent splicing (right blot; see Supplementary Fig. 2). Black arrows indicate band positions of the respective constructs (Actual MW of constructs: WT, 227 kDa; C-construct, 79 kDa; C + X, 65 kDa; C-terminal cleavage product, C*, 58 kDa). Note that X and X[REC] refer to $X_{Nav1.5}^{REC}$ in this panel. **d** Steady-state inactivation and conductance–voltage (G–V) relationships for functional constructs tested. **e** Comparison of values for half-maximal (in)activation ($V_{50}$) (values are displayed as mean +/- SD; WT, $n = 12$; N1472C, $n = 14$; N + C + $X_{Nav1.5}^{REC}$, $n = 16$). Significant differences were determined by one-way ANOVA with a Tukey post-hoc test. *$p < 0.03$ (WT vs. N1472C, $p = 0.014$; WT vs. N + C + $X_{Nav1.5}^{REC}$, $p = 0.012$). Source data are provided as a Source data file.

terminal amino acids 1503–2016 of $Na_v1.5$ (equivalent to DIV); and (3) a peptide X fragment (termed $X_{Nav1.5}^{REC}$) corresponding to the sequence to be replaced in the DIII–DIV linker of $Na_V1.5$ (amino acids 1472 to 1502 of $Na_V1.5$ but with N1472 mutated to Cys to enable splicing) flanked N- and C-terminally by Int[C] of *Cfa*DnaE (Int[C]-A) and Int[N] of *Ssp*DnaB[M86] (Int[N]-B), respectively (Fig. 2a). These constructs were transcribed into messenger RNA (mRNA) and injected into *Xenopus laevis* oocytes for recombinant expression. This approach is well-established for assessing ion channel function using electrophysiology and, conveniently, allows for direct delivery of mRNA and/or peptides into the cytosol using microinjection[26].

As the peptide X fragment contained the N1472C mutation, we first compared the function of WT channels with N1472C mutant channels and the spliced product resulting from co-injection of N + C + $X_{Nav1.5}^{REC}$ (Fig. 2b). As expected, injection of full-length WT and N1472C mRNA constructs resulted in robust channel expression, although the steady-state inactivation profile of

N1472C was shifted slightly to more depolarized potentials, consistent with earlier reports, suggesting for the N1472 locus to be potentially involved in cardiac disease[27] (Fig. 2b–e and Supplementary Table 1). Remarkably, co-injection of mRNA corresponding to N + C + $X_{Nav1.5}^{REC}$ (i.e., containing the N1472C mutation) resulted in full-length channels that showed robust current levels and were functionally indistinguishable from the full-length, recombinantly expressed channel construct also bearing the N1472C mutation (Fig. 2d, e). Importantly, co-expression of only two of the three constructs (i.e., N + C, N + $X_{Nav1.5}^{REC}$, or C + $X_{Nav1.5}^{REC}$) did not result in any voltage-dependent sodium current (Fig. 2b). Immunoblot analysis of co-expressed proteins also verified the presence of fully spliced $Na_v1.5$ when $X_{Nav1.5}^{REC}$ was co-expressed with both N and C constructs, although the relative abundance of fully spliced product was low compared to unspliced or splicing side products (<2% estimated based on immunoblots of total cell lysates; Fig. 2c). Importantly, a band corresponding to fully spliced

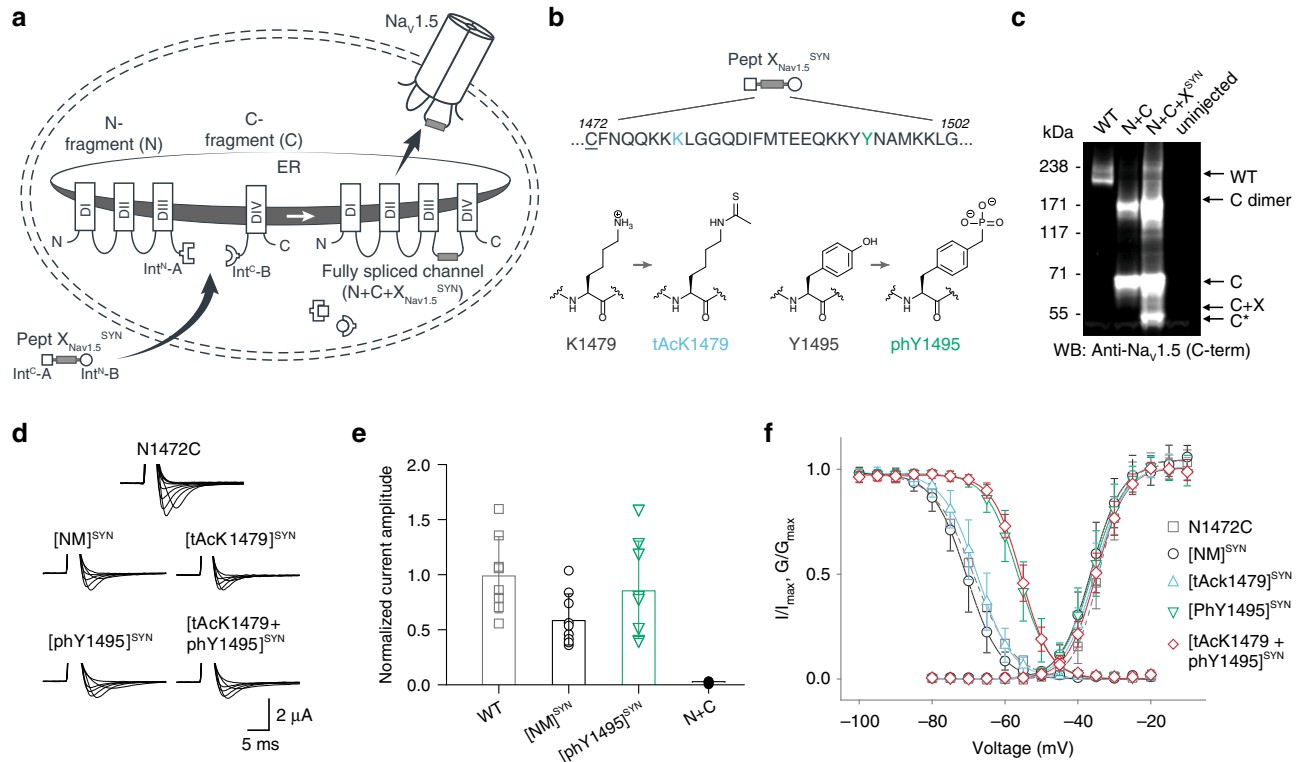

**Fig. 3 Insertion of synthetic peptides into NaV1.5. a** Schematic of strategy to reconstruct full-length Na$_v$1.5 from recombinantly expressed N-/C-terminal fragments (N and C) and a synthetic peptide (X$_{Nav1.5}$$^{SYN}$) in *Xenopus laevis* oocytes. Inteins A (*Cfa*DnaE) and B (*Ssp*DnaB$^{M86}$) indicated by square and round symbols, respectively. Note: we cannot exclude that splicing takes place at a different subcellular location than depicted. **b** Peptide X$_{Nav1.5}$$^{SYN}$ sequence corresponding to amino acids replaced in the Na$_v$1.5 DIII-DIV linker and chemical structures of native amino acids and PTM mimics (tAcK/phY) incorporated via chemical synthesis of peptide X$_{Nav1.5}$$^{SYN}$. The N1472C (underlined) mutation was introduced to optimize splicing (Supplementary Fig. 1). **c** Immunoblot verifying the presence of fully spliced Na$_v$1.5 only when peptide X$_{Nav1.5}$$^{SYN}$ was co-injected with N and C. Black arrows indicate band positions of the respective constructs (Actual MW of constructs: WT, 227 kDa; C, 79 kDa; C + X, 65 kDa; C-terminal cleavage product, C*, 58 kDa). Band at ~150 kDa is possibly a dimer or aggregate of C. **d** Representative sodium currents (see Methods; only voltage steps from −50 to +10 mV in 10 mV steps are displayed), demonstrating expression of functional Na$_v$1.5 when *Xenopus laevis* oocytes expressing N and C constructs were injected with synthetic peptides containing non-modifiable side chains in positions 1479 and 1495 (K1479R and Y1495F, NM), tAcK1479 or phY1495 or both PTM mimics together. **e** Average current amplitudes recorded at −35 mV from oocytes expressing N and C constructs and injected with synthetic peptide variant NM or phY1495 depicted as a bar plot (mean +/− SD; WT, $n = 8$; NM, $n = 9$; phY, $n = 8$; N + C, $n = 6$). Currents normalized to mean currents measured from oocytes expressing the full-length WT construct. To ensure adequate control of voltage clamp, [Na$^+$] in the extracellular recording solution was reduced (see Supplementary Fig. 3 for details). **f** Steady-state inactivation and conductance–voltage (G–V) relationships for PTM-modified/non-modified constructs (values displayed as mean +/− SD; N1472C, $n = 15$; NM, $n = 10$; tAcK1479, $n = 21$; phY1495, $n = 19$; tAcK1479 + phY1495, $n = 14$). Source data are provided as a Source data file.

product was not detected when a splicing-incompetent mutation (+1 extein Ser to Ala mutation in the C construct at splice site B) was introduced (N + C + X$_{Nav1.5}$$^{REC}$(mut.) in Fig. 2c). Indeed, non-covalent assembly arising from split intein cleavage products and/or partially spliced channel fragments did not occur within the typical timeframe of our experiments (Supplementary Fig. 1). Altogether, these data demonstrate that tPTS can be used to assemble full-length Na$_v$1.5 in live cells.

Having established that recombinant expression of N + C + X$_{Nav1.5}$$^{REC}$ can yield functional Na$_v$1.5 channels, we next generated synthetic versions of peptide X (X$_{Nav1.5}$$^{SYN}$; see Supplementary Fig. 2 for synthesis strategy) for injection into cells expressing only the N and C fragments recombinantly (Fig. 3a). Specifically, we synthesized X$_{Nav1.5}$$^{SYN}$ constructs that contained one of the following four variants: (i) mutations K1479R and Y1495F (termed [NM]$^{Syn}$) to prevent acetylation and phosphorylation, respectively; (ii) a thio-acetylated Lys analog at position 1479 (tAcK1479) that mimics PTM but displays increased metabolic stability against sirtuins compared to regular acetylation[28,29]; (iii) a phosphonylated Tyr analog at

position 1495 (phY1495) that provides a non-hydrolysable phosphate mimic; or (iv) both tAcK1479 and phY1495 to mimic a dual PTM scenario (Fig. 3b). The N and C fragments were recombinantly expressed in oocytes for 24 h before injection of the synthetic X$_{Nav1.5}$$^{SYN}$ variants. Successful splicing of full-length Na$_v$1.5 containing one of the four synthetic X$_{Nav1.5}$$^{SYN}$ variants was verified by immunoblotting and electrophysiology (Fig. 3c, d). As before, the relative abundance of fully spliced product estimated from immunoblot analysis was low compared to the abundance of unspliced or splicing side products (<1% in total cell lysates), but expression of robust voltage-gated sodium currents was achieved within 12 h of X$_{Nav1.5}$$^{SYN}$ variant injection. In fact, observed current levels at 24 h post peptide injection (i.e., 48 h after injection of N- and C-mRNA) were comparable to those observed 48 h after injection of WT mRNA (Fig. 3e and Supplementary Fig. 3). To the best of our knowledge, this represents the first incorporation of a tAcK residue and the first insertion of two distinct PTM mimics in a full-length protein in eukaryotic cells. Functional analysis demonstrated that the voltage dependence of activation was not affected by any of the

introduced PTM mimics or the conventional K1479R and Y1495F mutations. Conversely, insertion of phY1495, either alone or in combination with tAcK1479, induced a clearly discernable (15 mV) rightward-shift in the voltage-dependence of fast inactivation (Fig. 3f and Supplementary Table 1). These data are consistent with earlier reports, suggesting that acetylation of K1479 primarily affects current density[24], whereas phosphorylation of Y1495 affects fast inactivation properties[25].

To further validate our approach and demonstrate its suitability for other target sequences, we applied the same strategy to the intracellular linker connecting DI and II of Na$_V$1.5. Similar to the DIII–IV linker, mutations or aberrant PTMs in this region of Na$_V$1.5 have been implicated in cardiac disease[23,30]. Using appropriate N and C constructs, together with both recombinantly expressed and synthetic versions of a peptide X$_{Nav1.5}$ variant corresponding to amino acids 505 to 527 of Na$_V$1.5, we demonstrated that tPTS can be used to probe the function of different intracellular regions of Na$_V$1.5 in *Xenopus* oocytes. Specifically, we found that neither methylation of R513 (meR513), nor phosphonylation of S516 (phS516), nor their combined presence[31], affected activation or inactivation of Na$_V$1.5 (Supplementary Figs. 4 and 5 and Supplementary Table 1).

**Semi-synthesis of GFP in HEK cells.** The above data showed that tPTS could be used to insert synthetic peptides into large membrane proteins in live eukaryotic cells, but it was important to demonstrate delivery into mammalian cells, which can be more challenging. To demonstrate the feasibility of this approach in mammalian cells, we split GFP into three fragments (analogous to our approach with Na$_v$1.5 described above): (1) an N-terminal construct (N-GFP) corresponding to amino acids 1–64 of GFP, fused to Int$^N$ of *Cfa*DnaE; (2) a C-terminal construct (C-GFP) corresponding to Int$^C$ of *Ssp*DnaB$^{M86}$ linked to amino acids 86–238 of GFP, and (3) a peptide X fragment (X$_{GFP}^{REC}$) corresponding to amino acids 65 to 85 and flanked by Int$^C$ of *Cfa*DnaE at the N-terminus and by Int$^N$ of *Ssp*DnaB$^{M86}$ at the C-terminus (Fig. 4a). The constructs were co-expressed in different combinations in human embryonic kidney (HEK) cells, which expressed functional GFP only when all three constructs (N-GFP + C-GFP + X$_{GFP}^{REC}$) were transfected, albeit with low yields (~4%, as estimated by fluorescence-activated cell sorting (FACS), Supplementary Fig. 6). No GFP fluorescence was detected with the co-expression of any two constructs or when cells were transfected with constructs containing splicing-incompetent mutations (C65A at +1 extein X$_{GFP}^{REC}$ (splice site A) or S85A at +1 extein C-GFP (splice site B); Fig. 4b).

We subsequently sought to generate a semi-synthetic GFP by delivering synthetic peptide X$_{GFP}^{SYN}$ variants into HEK cells that recombinantly expressed N- and C-terminal fragments of GFP (Fig. 5a). We achieved delivery of synthetic peptides using the transient cell permeabilization method known as cell squeezing, which involves rapid viscoelastic deformation[32]. Although yields were low (~1%), GFP fluorescence was detected only in N-GFP- and C-GFP-transfected cells that had been squeezed in the presence of peptide X$_{GFP}^{SYN}$ (Fig. 5b). The approach further allowed us to incorporate the ncAA 3-nitro-tyrosine at position 66 of GFP to replace the tyrosine that is involved in chromophore formation. This modification resulted in a blue-shift in the spectral properties of GFP and confirmed the utility of tPTS for creating semi-synthetic variants in mammalian cells (Supplementary Fig. 7).

**Insertion of ncAAs into P2X2 receptor extracellular domain.** While standard PTS has been employed to splice numerous

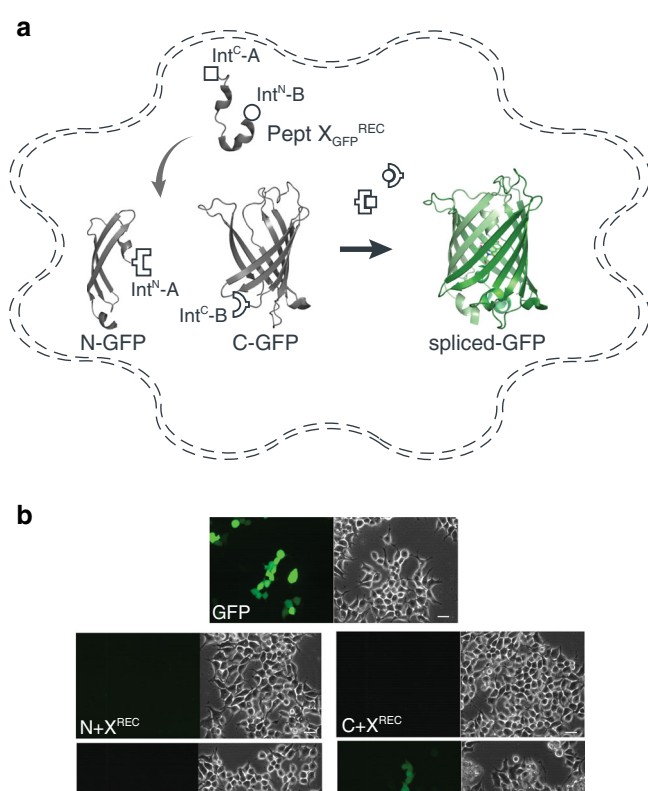

**Fig. 4 Insertion of a recombinantly expressed peptide into GFP expressed in mammalian cells. a** Schematic presentation of the strategy applied to reconstitute GFP from recombinantly expressed N-/C-terminal fragments and recombinantly expressed peptide X$_{GFP}^{REC}$ corresponding to amino acids 65–85 of GFP in HEK293 cells. Inteins A (*Cfa*DnaE) and B (*Ssp*DnaB$^{M86}$) are indicated by square and round symbols, respectively. **b** Bright-field (right panels) and fluorescence (left panels) images of HEK293 cells expressing the indicated constructs. Scale bars: 20 μm. GFP fluorescence was only detected when all three constructs (N + X + C) were co-transfected. GFP fluorescence was not detected when one of the three constructs was absent or when +1 extein residues of each split intein is mutated to alanine (C65A for intein A and S86A for intein B) to prevent splicing.

cytosolic proteins and peptides, extracellular targets are more challenging and have been rarely investigated using PTS[17]. We sought to test whether tPTS could be used to insert synthetic peptides into an extracellular protein domain. We chose the P2X2 receptor (P2X2R), a trimeric ATP-gated ion channel whose extracellular domain binds ATP released during synaptic transmission[33]. While the location of the ATP-binding site in the extracellular domain is undisputed, the details of how conserved basic side chains coordinate the phosphate tail of ATP remain unclear[34]. However, ribosome-based non-sense suppression approaches, using, e.g., ncAA analogs of lysine, have failed at position K71 in the P2X2R, likely due to non-specific incorporation of endogenous amino acids (Supplementary Fig. 8). We, therefore, used tPTS to test whether the charge position of K71 is crucial for ATP recognition.

As an initial proof-of-concept of splicing within an extracellular domain of a membrane receptor, we used standard PTS to

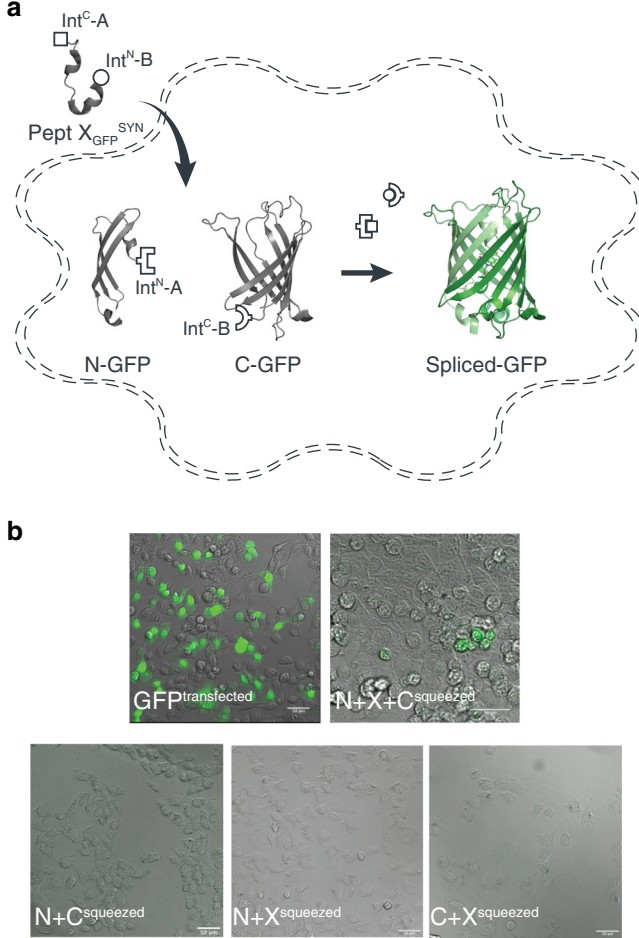

**Fig. 5 Insertion of synthetic peptides into GFP. a** Schematic of strategy applied to reconstitute GFP from recombinantly expressed N-/C-terminal fragments and a synthetic peptide $X_{GFP}^{SYN}$ in HEK293 cells. **b** Overlay of bright-field and fluorescence images taken of HEK293 cells transfected with only WT GFP, or N and C fragments and squeezed in the presence or absence of peptide $X_{GFP}^{SYN}$. Scale bars: 50 μm.

confirmed that splicing was highly efficient, with near complete conversion of the N and C fragments to full-length receptors.

We proceeded to test splice site B, employing an analogous approach to that implemented for splice site A (Supplementary Fig. 10a, b), except we used the $Ssp$DnaB$^{M86}$ split intein in this case together with amino acids 1–75 of P2X2 (N-terminal fragment) and amino acids 76–472 of P2X2 plus a $faux$ transmembrane segment (C-terminal fragment). Similar to the results obtained at splice site A, full-length P2X2 receptors with WT-like ATP sensitivity were only present upon co-expression of N + C, but not N or C alone (Supplementary Fig. 10c–e). Although lower than in the case of splice site A, the splicing was still efficient, with over half of the N and C fragments being converted into full-length receptors (Supplementary Fig. 10d). In contrast to splice site A, co-expression of the N and splicing-deficient C (S76A) constructs resulted in small ATP-gated currents in response to high concentrations of ATP ( > 1 mM) after long incubation times (>48 h). Of note, the prevention of splicing favors side reactions, which will result in the accumulation of cleavage products. It is thus possible that the non-covalent assembly of the N and C cleavage products results in a receptor population with impaired function, as evident from the drastically reduced ATP sensitivity we observed (Supplementary Fig. 10e). However, this result likely overstates the likelihood of cleavage products occurring compared to when splicing-competent split inteins are used. Overall, these data confirm that splicing can be achieved within an extracellular domain of a membrane receptor.

In order to insert a peptide fragment into the extracellular domain of P2X2 using tPTS, we used an analogous approach to that described for Na$_v$1.5 and GFP to generate three constructs: (1) an N-terminal construct (N) corresponding to amino acids 1–53 of P2X2 fused to Int$^N$ of $Cfa$DnaE; (2) a C-terminal construct (C) containing a $faux$ transmembrane domain linked to Int$^C$ of $Ssp$DnaB$^{M86}$ and amino acids 76–472 of P2X2; and (3) a peptide X fragment (termed $X_{P2X2}^{REC}$) containing amino acids 54 to 75 of P2X2 flanked N- and C-terminally by Int$^C$ of $Cfa$DnaE and Int$^N$ of $Ssp$DnaB$^{M86}$, respectively (Fig. 6a). To optimize splicing efficiency, we additionally tested a C-terminal construct with a cleavable $faux$ transmembrane domain, which comprised an IgK N-term signal sequence and a signal peptidase cleavage site inserted between the $faux$ transmembrane segment and Int$^C$ of $Ssp$DnaB$^{M86}$. The resultant current amplitudes confirmed superior performance compared to the non-cleavable $faux$ domain (Supplementary Fig. 11), therefore further experiments proceeded with this optimized C-terminal construct. Following expression in $Xenopus$ $laevis$ oocytes, splicing of full-length, ATP-gated receptors was only apparent when all three fragments (N, C, and $X_{P2X2}^{REC}$) were present, and represented an estimated 7% of the total products/reactants detected by immunoblotting (Fig. 6b, c). Importantly, introduction of the S76A mutation at the +1 extein position of the C construct did not result in detectable currents. Further, introduction of the K71Q mutation-bearing peptide $X_{P2X2}^{REC}$ into the spliced receptors shifted the ATP concentration-response curve to the right to a similar degree as the conventional K71Q mutant (Fig. 6d). Altogether, these data demonstrate successful and splicing-dependent assembly of functional P2X2 receptors upon co-expression of N, C, and $X_{P2X2}^{REC}$ constructs.

Finally, to test whether the position of the charge at K71 is crucial for ATP recognition, we synthesized peptide $X_{P2X2}^{SYN}$ variants containing lysine and ncAA lysine derivatives (homolysine, hLys, and ornithine, Orn) at position 71 (Fig. 6e), which differed only in the length of their side chains. Following recombinant expression of N and C in $Xenopus$ $laevis$ oocytes and injection of synthetic peptide, successful splicing was confirmed by functional responses to ATP application (Supplementary Fig.

independently assess splicing at either side of K71 in P2X2R: S54, which was mutated to Cys to improve splicing (splice site A), and S76 (splice site B). Again, we chose $Xenopus$ $leavis$ oocytes as an expression system, as they allow facile peptide delivery. For splice site A, the N-terminal fragment contained amino acids 1-53 of P2X2 linked to Int$^N$ of $Cfa$DnaE. However, the C-terminal construct contained a $faux$ transmembrane domain (amino acids 1–74 from ASIC1a) followed by Int$^C$ of $Cfa$DnaE and the C-terminal receptor fragment of P2X2 (amino acids 54–472) (Supplementary Fig. 9a). Introduction of the $faux$ transmembrane segment was necessary to enforce the correct topology of the resulting construct. To demonstrate that successful splicing is necessary for assembly of full-length receptors, we also generated a version of the C-terminal construct containing the C54A mutation, which effectively removes the required +1 Cys side chain and renders the construct splicing incompetent (Supplementary Fig. 9b). Expression of the individual constructs alone (N or C) in $Xenopus$ oocytes did not result in functional receptors, whereas co-expression of N + C (but not N + C (C54A)) resulted in receptors with WT-like ATP sensitivity (Supplementary Fig. 9c, d). Confirmation of correct splicing was provided by immunoblots showing that bands corresponding to WT P2X2 only occurred in the presence of N + C, but not any of the control constructs (Supplementary Fig. 9e). Of note, biochemical analysis

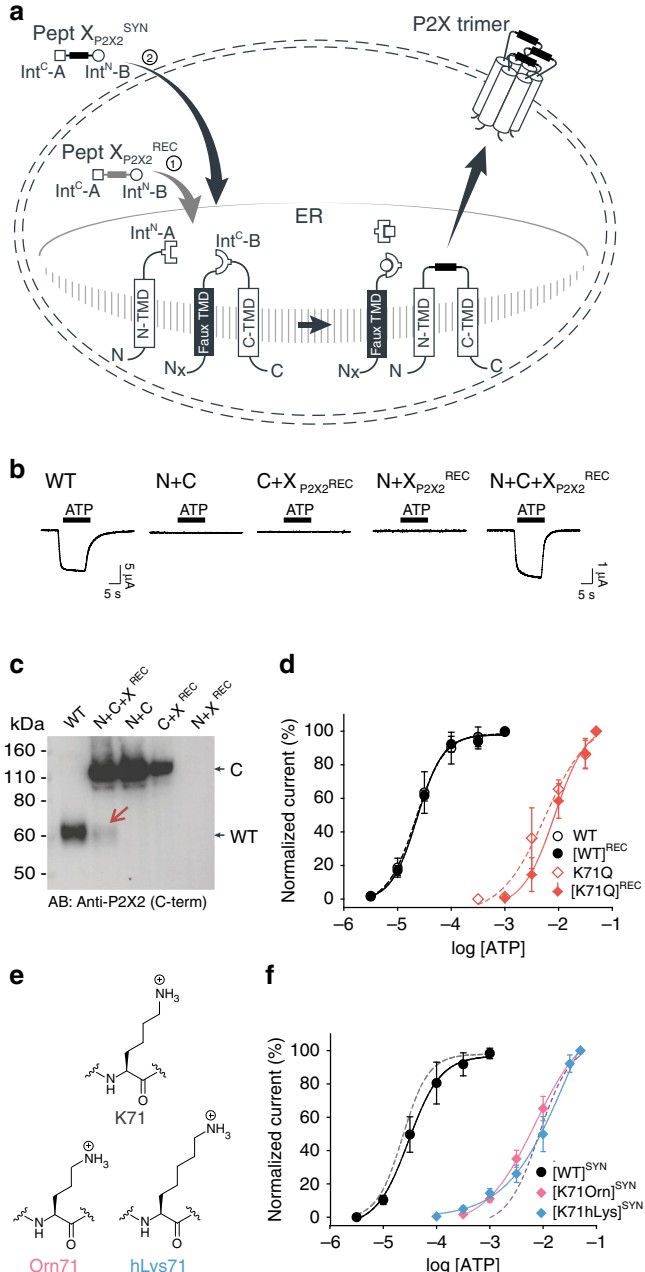

**Fig. 6 Insertion of recombinant and synthetic peptides into the P2X2R extracellular domain. a** Schematic presentation of the strategy to reconstruct full-length P2X2Rs from recombinantly expressed N-/C-terminal fragments (N and C) and a recombinant (strategy 1, Pept $X_{P2X2}^{REC}$) or synthetic peptide (strategy 2, Pept $X_{P2X2}^{SYN}$) into the P2X2R extracellular domain in *Xenopus laevis* oocytes. Note that a *faux* transmembrane helix (*faux* TMD) was engineered into the C-terminal fragment to maintain its native membrane topology (see also Supplementary Fig. 9). Inteins A (*Cfa*DnaE) and B (*Ssp*DnaB^M86^) indicated by square and round symbols, respectively. Peptide X was designed to include a C-terminal ER-targeting KDEL signal sequence, which is excised during the splicing process. However, we cannot exclude the possibility that splicing takes place at a different subcellular location than depicted here. **b** Representative current traces obtained by application of 30 μM ATP, verifying functional expression only when all three components were recombinantly co-expressed (strategy 1 in **a**). **c** Immunoblot of surface-purified proteins. Black arrows on the right indicate band positions of the respective constructs (actual MW of constructs: WT, 53 kDa; C-construct, 97 kDa). Red arrow indicates the band of the spliced full-length receptor. **d** ATP concentration-response curves (CRCs) for P2X2 WT (black) and K71Q (red) reconstituted from the three co-expressed constructs. Dashed lines represent the CRCs for the respective full-length proteins. **e** Structures of inserted side chains at position 71 (Lys, Orn, hLys) incorporated via synthetic peptide X (strategy 2 in **a**). **f** ATP CRCs indicate successful incorporation of synthetic WT peptide $X_{P2X2}^{SYN}$ (black) with wild-type like ATP sensitivity, while synthetic peptides with Orn (pink) or hLys (blue) resulted in a decrease in ATP sensitivity. Dashed lines indicate ATP CRCs for spliced P2X2R WT (gray) or K71Q (purple) obtained with peptide $X_{P2X2}^{REC}$. Values in **d** and **f** are displayed as mean +/– SD; WT, $n = 5$; [WT]^REC^, $n = 8$; K71Q, $n = 7$; [K71Q]^REC^, $n = 9$; [WT]^SYN^, $n = 7$; [K71Orn]^SYN^, $n = 9$; [K71hLys]^SYN^, $n = 5$. Source data are provided as a Source data file.

P2X2R. Application of the tPTS approach to incorporate hLys and Orn at position 69 (replacing a different lysine residue involved in ATP recognition) resulted in currents not distinguishable from background (Supplementary Fig. 12). This suggested that the modification resulted in an even larger right-shift in the ATP concentration-response curve, which cannot be accurately determined. However, despite this low overall splicing efficiency, we were able to use conventional PTS to reconstitute functional P2X2Rs from N- and C-terminal fragments expressed in HEK cells using only a single (*Cfa*DnaE) split intein (Supplementary Fig. 13), demonstrating that splicing within extracellular domains is feasible in both *Xenopus laevis* oocytes and mammalian cells.

## Discussion

We have demonstrated that tPTS can be employed to introduce single or multiple chemical modifications into soluble and membrane proteins in live cells. This includes combinations of ncAAs, PTMs or PTM mimics that cannot currently be incorporated into live cells using available methods. A key advantage of the tPTS approach in live cells is that the refolding step typically required with in vitro applications can be bypassed. This means the approach can be used for larger, more complex proteins, including those residing in the membrane. Additionally, the approach does not rely on the ribosomal machinery and thus delivers a homogenous protein population by avoiding the potential for non-specific incorporation, which can affect protein manipulation using non-sense suppression approaches[32–36].

While tPTS offers unique ways to manipulate proteins, several aspects require careful consideration for its applications in a broader context. First, the splicing efficiency in tPTS is sequence-

12 and Fig. 6f). The ATP sensitivity of these responses demonstrated that the Lys-containing peptide $X_{P2X2}^{SYN}$ variant generated WT-like responses, whereas those containing hLys and Orn generated responses similar to those obtained with the conventional K71Q mutant (Fig. 6f). We thus conclude that efficient recognition of ATP by P2X2Rs is highly dependent on the precise position of the charge at K71. However, ATP-generated currents were markedly smaller (<5%) than those recorded from full-length protein (WT) or from the spliced product generated by co-expression of $X_{P2X2}^{REC}$ with N and C, even after attempts to increase the concentration of peptide $X_{P2X2}^{SYN}$ inserted into oocytes using multiple injections (see Methods). Additionally, functional currents took a longer time to manifest (3–5 days after peptide $X_{P2X2}^{SYN}$ injection, compared to 1 day after WT and 3 days after N + C + $X_{P2X2}^{REC}$ RNA injection), indicating slow formation of the fully spliced product. This low splicing efficiency was also evident from our inability to use immunoblotting to detect bands corresponding to full-length

dependent. In cases where the native sequence does not contain residues required for splicing, mutations may need to be introduced at the intended splice sites to fulfill the extein requirements for successful splicing (i.e., the need for a Cys or Ser at the +1 extein position of the extein, see Fig. 1). Moreover, the protein fragment to be modified needs to be within the length limit of what is synthetically feasible. We also expect for example transmembrane sections of a protein to be less amenable to this method, as they are challenging to synthesize and insert post-translationally. Second, we note that numerous other split inteins[35] with different extein requirements could alternatively be used for this approach and could potentially, depending on the context, yield higher splicing efficiency. Here, we chose the split inteins $Cfa$DnaE and $Ssp$DnaB$^{M86}$, as they have been well-characterized with fast kinetics and engineered to have increased tolerance to non-native extein sequences[10,11]. Importantly, $Ssp$DnaB$^{M86}$ can be split asymmetrically with the Int$^N$ segment only comprising 11 amino acids, making it an ideal split-intein B in this approach (Fig. 1). Finally, the means of introducing the synthetic peptide X needs to be optimized depending on the cell type in question. While synthetic peptides can be injected directly into $Xenopus\ laevis$ oocytes, our approach requires potentially more challenging delivery techniques, such as cell squeezing, electroporation or the use of cell-penetrating peptides when implemented in mammalian cells.

Unsurprisingly, for all the proteins tested here, we note that the amount of fully spliced products generated using tPTS is generally lower, and their formation can take longer than when expressing full-length WT proteins. Factors such as molecular crowding or unfavorable spatial arrangements of protein fragments in the cell could contribute to these issues. Furthermore, it cannot be excluded that the recombinantly expressed protein fragments display different stabilities toward the proteasome or are differentially trafficked, resulting in unequal fragment ratios, and thus potentially suboptimal conditions for splicing to occur. The length, proteolytic stability, and solubility of synthetic peptides, along with requirements for native-like flanking extein sequences, can also affect splicing efficiency and reaction rates[36]. Additionally, the amount of synthetic peptide that can be delivered into a cell is typically limited by the viability of the cell in response to delivery of the peptide and peptide concentration. Lastly, factors that contribute to optimal splicing conditions, such as pH or redox potential, which are controllable in vitro, are virtually impossible to manipulate in a live cell. Indeed, it is possible that the lower splicing efficiency we observed when using tPTS to modify the extracellular domain of the P2X2 receptor was due to unfavorable redox conditions in the endoplasmic reticulum and/or the low abundance of synthetic peptide in this sub-cellular compartment (or others that the splicing could take place in).

Nevertheless, it is important to appreciate that low protein yields are also not uncommon with ribosome-based approaches to genetically engineer proteins. This is particularly true for complex proteins expressed in eukaryotic cells. In fact, many groups have repeatedly observed yields of 10% or less with ncAA incorporation into transporters[37], ion channels[38–40], and G protein-coupled receptors[41,42]. Although the generally low yields observed with tPTS likely restrict the approach to applications that do not require large amounts of protein, at least some of the above limitations can be addressed by engineering more promiscuous and efficient split inteins[10–12,43] or by adding affinity tags to promote split intein interactions[18]. Such improvements would allow the approach to be applied to a broader complement of target proteins.

The ability to apply this approach in eukaryotic cells has enabled us to use highly sensitive electrophysiology and imaging

techniques to determine the presence and functionality of fully spliced products. tPTS will thus permit synthetic peptide insertion into different proteins, in particular those that are amenable to highly sensitive methods to study function or localization. Beyond the introduction of PTMs, PTM mimics, and ncAAs, the approach can be used to insert virtually any chemical modification into a target protein, including backbone modifications, chemical handles, fluorescent or spectroscopic labels, and combinations thereof. This constitutes an important advantage over existing methodologies. Specifically, we anticipate that the approach will overcome some of the drawbacks associated with conventional genetic engineering in eukaryotic cells (non-specific incorporation, premature termination, dependence on ribosomal promiscuity[37,44]) and semi-synthetic approaches that require protein refolding[7]. It will thus increase the number and type of functionalities that can be incorporated into proteins that prove amenable to tPTS.

## Methods

**Molecular biology.** Plasmid DNAs were purchased from GeneArt (Thermo Fisher scientific), General biosystems Inc. or Twist Bioscience. All gene constructs were sub-cloned into either the pUNIV or pcDNA3.1+ backbone. pUNIV backbone was a gift from Cynthia Czajkowski (Addgene plasmid # 24705; http://n2t.net/addgene:24705; RRID:Addgene_24705). Conventional site-directed mutagenesis was performed using standard PCR. Primer sequences are listed in Supplementary Table 2. Complementary RNA (cRNA) for oocyte microinjection was transcribed from respective linearized cDNA using the Ambion mMESSAGE mMACHINE T7 Transcription Kit (Thermo Fisher Scientific).

**Peptide synthesis.** Peptides for GFP splicing were sourced from Proteogenix, France. Peptides for Na$_V$1.5 and P2X2R splicing were synthesized by solid-phase peptide synthesis (details in Supplementary material). Peptide X variants were synthesized as three shorter fragments and assembled in a one-pot native chemical ligation procedure, as briefly outlined below. The split intein-mediated reconstitution of proteins developed here required the synthesis of a small collection of peptides between 69 and 77 amino acids in length. Conveniently, all peptide X variants needed for our work share identical Int$^C$-A (35 amino acids) and Int$^N$-B (11 amino acids) sequences, which flank the sequence corresponding to the protein of interest (POI). In order to reduce the synthesis demands, we took advantage of the sequences of Int$^C$ and Int$^N$ (i.e., Cys residues at +1 position in the exteins) by adopting a "one-pot" chemical ligation strategy of three parts (Int$^C$-A, POI segment and Int$^N$-B), with the sequence from the POI being the only variable one. For this purpose, a C-to-N-directed ligation strategy based on Thz masking of cysteine[45] was implemented for the assembly of the peptide X variants (Supplementary Fig. 2). For the assembly of peptide X variants containing a thio-acetylated lysine, a different ligation strategy (N-to-C directed) was adopted (Supplementary Fig. 2) in order to avoid the Thz-cysteine unmasking step (acidic pH at 37 °C) of the C-to-N-directed ligation. Indeed, our collaborators experienced partial conversion of similar thioamide-containing peptides, likely into oxoamides, during work-up and purification (Dr. Christian A. Olsen, personal communication).

**Expression in $Xenopus\ laevis$ oocytes.** Stage V/VI oocytes were obtained from ovaries of female $Xenopus\ laevis$ frogs (anaesthetized in 0.3% tricaine, under license 2014-15-0201-00031, approved by the Danish Veterinary and Food Administration). cRNAs were injected into the oocytes (follicle layers enzymatically removed by 0.5 mg/mL Type I collagenase[39]) and incubated at 18 °C in OR-3 solution (50% Leibovitz's medium, 1 mM L-Glutamine, 250 mg/L Gentamycin, 15 mM HEPES, pH 7.6) for up to 7 days. For injection of synthetic peptides, lyophilized peptides were dissolved in Milli-Q H$_2$O to a concentration of 750 μM and 18 nL of solubilized peptide was injected into cRNA pre-injected oocytes with the $Nanoliter\ 2010$ micromanipulator (World Precision Instruments). For Na$_V$1.5 constructs, synthetic peptides were injected 1 day after cRNAs were injected and recordings performed 12–20 h after peptide injection. For P2X2 constructs, synthetic peptides were injected consecutively on days 2, 3, and 4 following cRNA injection and recordings performed on day 7.

**Two-electrode voltage clamp (TEVC) recordings.** Voltage or ATP-induced currents were recorded with two microelectrodes using an OC-725C voltage clamp amplifier (Warner Instruments). Oocytes were perfused in ND96 solution (in mM: 96 NaCl, 2 KCl, 1 MgCl$_2$, 1.8 CaCl$_2$/BaCl$_2$, 5 HEPES, pH 7.4) during recordings. Glass microelectrodes were backfilled with 3 M KCl and microelectrodes with resistances between 0.2 and 1 MΩ were used. Oocytes were held at −100 mV (for Na$_V$1.5 constructs) or −40 mV (for P2X2 constructs). For Na$_V$1.5, sodium currents were induced by +5 mV voltage steps from −80 to +40 mV. Steady-state inactivation was measured by delivering a 500 ms prepulse from −100 to −20 mV in +5

mV voltage steps followed by a 25 ms test pulse of −20 mV. For P2X2 recordings, ATP-induced currents were elicited through application of increasing concentrations of ATP (dissolved in ND96, pH 7.4) supplied via an automated perfusion system operated by a ValveBank™ module (AutoMate Scientific).

**Immunoblots.** Oocytes expressing full-length receptors or different combinations of the split-intein receptor fragment fusion proteins were isolated 3–4 days after RNA injection and washed twice with phosphate-buffered saline (PBS). Total cell lysates were obtained by lysing the oocytes in Pierce™ IP lysis buffer with added Halt protease inhibitor cocktail (Thermo Fisher Scientific). Surface proteins were purified with the Pierce™ Cell Surface Protein Isolation Kit (Thermo Fisher Scientific). Purified surface proteins or total cell lysates were run on a 4–12% BIS-TRIS gel (for P2X2) or 3–8% Tris-acetate gel (for $Na_V1.5$) and transferred to a polyvinylidene difluoride membrane. Membranes were incubated with rabbit polyclonal anti-$Na_V1.5$ (#ASC-005, Alomone labs; 1:2000), anti-$Na_V1.5$ (#ASC-013, Alomone labs; 1:1500) or anti-P2X2 Antibody (#APR-003, Alomone labs; 1:2000) and the bound primary antibodies were detected by a HRP-conjugated goat anti-rabbit secondary antibody (W401B, Promega; 1:2000). Membranes were developed and visualized using the Pierce™ ECL immunoblotting substrate (Thermo Fisher Scientific).

**Expression in HEK293 cells.** HEK293 cells (American Type Culture Collection) were grown in Dulbecco's modified Eagle's Medium (DMEM) (Gibco) supplemented with 10% fetal bovine serum (Gibco), 100 units/mL penicillin and 100 μg/mL streptomycin (Gibco) and incubated at 37 °C with 5% of $CO_2$. Confluent cells growing in monolayers were washed with 10 mL PBS (in mM: 137 NaCl, 2.7 KCl, 4.3 $Na_2HPO_4$, 1.4 $KH_2PO_4$ (pH 7.3)), detached with trypsin-EDTA (Thermo Fisher Scientific) and re-suspended in DMEM. The re-suspended cells were seeded onto glass coverslips pre-treated with poly-L-Lysine in 35-mm dishes for patching or in 35-mm glass bottom dishes for imaging and incubated for 24 h, prior transfection. The plated HEK293 cells were transfected using TransIT DNA transfection reagent (Mirus) following the instructions supplied by the manufacturer and incubated until use. For imaging of reconstituted GFP in HEK293 cells, DNA coding for three GFP-split-intein fusion fragments (N, X, and C) was inserted into the pcDNA3.1+ vector by GeneArt (Thermo Fisher Scientific) and co-transfected in a 1:1:1 ratio using a total of 3 μg DNA and incubated for 48 h. before imaging. In parallel, a batch of cells was transfected with WT GFP as a positive control and in addition five batches of cells were transfected with DNA coding for two fragments of the GFP alone (N + X, N + C, X + C) or combined with a non-splicing GFP fragment (N + X-Cys65Ala + C or N-X + C-Ser85Ala) as negative controls. To keep the same amount of DNA for each combination pcDNA3.1 + empty vector was co-transfected for the control experiments. For P2X2R patch-clamp recordings, HEK293 cells were transfected in a 30 mm dish with 1.5 μg DNA of each construct, respectively (N + C, N + $C_{mut}$, N, C) and incubated for 2 days at 37 °C.

**Imaging of reconstituted GFP.** Imaging was performed using an inverted microscope *IX73* (Olympus) with 10x and 20x objectives mounted on a motorized nosepiece (Olympus) controlled by a *CMB U-HSCBM* switch and connected to a *DCC1545-M* camera (ThorLabs). GFP fluorescence was visualized using a LED light source (CoolLed pE-100, 470 nm).

**Peptide transfer by cell squeezing.** Squeezing was performed using a chip with constrictions of 6 μm in diameter and 10 μm in length (CellSqueeze 10-(6)x1, SQZbiotech). In all microfluidic experiments, a cell density of $1.5 \times 10^6$ cells/mL in Opti-MeM was squeezed through the chip at a pressure of 40 psi. Transduction was conducted at 4 °C to block cargo uptake by endocytosis[46]. During squeezing, a peptide concentration of 10–20 μM in the surrounding buffer was used. After squeezing, cells were incubated for 5 min at 4 °C to reseal the plasma membrane. Squeezed cells were washed with DMEM containing 10% FCS, seeded into 8-well on cover glass II slides (Sarstedt) coated with fibronectin (5 μg/mL) in DMEM containing 10% FCS, and cultured at 37 °C and 5% $CO_2$. As a control for endosomal uptake, cells were incubated with 10 μM of peptide at room temperature without microfluidic cell manipulation. Confocal imaging was performed 1, 2, 4, 8, and 20 h after squeezing. Before imaging, cells were washed with PBS (Sigma-Aldrich), fixed with 4% formaldehyde (Roth)/PBS for 20 min at 20 °C and quenched by the addition 50 mM glycine in PBS (10 min, 20 °C).

**Confocal laser-scanning microscopy.** Imaging was performed using the confocal laser-scanning (LSM) microscope LSM880 (Zeiss) with Plan-Apochromat 20×/1.4 Oil DIC objective. The following laser lines were used for excitation: 405 nm for blue-shifted GFP and 488 nm for GFP. ImageJ[47], Fiji[48], and Zen 2.3 black (Carl Zeiss Jena GmbH, Germany) were used for image analysis.

**Patch-clamp electrophysiology.** The cells were reseeded 1 to 4 h before the patch-clamp experiments. Ionic currents were recorded with borosilicate patch pipettes (2–5 MΩ) filled with intracellular solution (in mM: 140 KCl, 5 $MgCl_2$, 5 EGTA, 10 HEPES, pH 7.3) at −40 mV with the Axopatch 200B amplifier and the 1550A digitizer (Molecular Devices). Lifted cells were perfused with extracellular solution (in mM: 140 NaCl, 2.8 KCl, 2 $CaCl_2$, 2 $MgCl_2$, 10 HEPES, 10 Glucose, pH 7.3) and activated with 1 mM ATP via a piezo-actuated perfusion tool.

**Fluorescence-activated cell sorting (FACS).** HEK293 cells were grown in Dulbecco's modified Eagle's Medium (DMEM) (Gibco) supplemented with 10% Fetal Bovine Serum (Biowest), and incubated at 37 °C with 5% of $CO_2$. Four-hundred thousand= cells were seeded in 6-well plates and incubated for 24 h, prior transfection. DNA coding for three GFP-split-intein fusion fragments (N, X, and C) was co-transfected in a 1:1:1 ratio using a total of 4.5 μg DNA. To keep the same amount of DNA for each combination pcDNA3.1+ empty vector was co-transfected for the N + C and WT eGFP control experiments. Cells were transfected using 6 μg PEI and incubated for circa 44 h. The cells were then detached with trypsin-EDTA, spun down and re-suspended in PBS containing formaldehyde 37% (1:40) and DAPI 200 μM (1:200). The cell suspension was then passed through a cell-strainer cap and analyzed with BD™ LSR II flow cytometer within 15 min.

**Statistics and reproducibility.** Electrophysiological data were recorded with pClamp10 (Molecular Devices) and analyzed in Clampfit 10 (Molecular Devices). Prism8 (Graphpad) and Sigmaplot 13.0 (SPSS) were used for statistical analysis. Values and error bars in all graphs represent the mean and standard deviation from at least five individual cells and two different batches, unless stated otherwise. Statistical differences were determined using one-way ANOVA with a Tukey post-hoc test, with $p > 0.03$ considered to be non-significant. Immunoblots and microscope images shown are representative of experiments from at least two batches of cells. FACS data was analyzed using FlowJo 10 (Becton Dickinson). Chemical structures were made in ChemDraw16 (PerkinElmer) and figures were prepared in Illustrator 2020 (Adobe).

**Reporting summary.** Further information on research design is available in the Nature Research Reporting Summary linked to this article.

## Data availability
The data supporting the findings of this study are available in the manuscript and supplementary files or are available from the corresponding author upon request. The source data underlying Figs. 2c–e, 3c, e–f, 6c, d, f, and Supplementary Figs. 1c, 3a, 4c–e, h–j, 5c, 8b, 9d–e, 10d–e, 11d are provided as a Source data file and are available on Zenodo https://doi.org/10.5281/zenodo.3712821 [https://zenodo.org/record/3712821#.Xp1pqP1KiUk] The pUNIV vector backbone was obtained from the Addgene repository (Addgene plasmid # 24705; http://n2t.net/addgene:24705; RRID:Addgene_24705).

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

## Acknowledgements

We acknowledge the Lundbeck Foundation (R139-2012-12390 to SAP), the Carlsberg Foundation (CF16-0504 to SAP), the Independent Research Fund Denmark (7025-00097A to SAP), the University of Copenhagen, and the German Research Foundation (SFB 807, SPP 1623, and GRK 1986 to R.T.) for financial support. R.T. would like to acknowledge the support by an ERC Advanced Grant from the European Research Council. We thank Dr. Christian A. Olsen for support with the peptide chemistry, Janne Colding and Natasha Gray-Garney for technical support, and Drs. Marlieke LM Jongsma and Huib Ovaa for help with the FACS experiments. We would also like to thank Drs. Lesley Anson, Christian A. Olsen, and Kristian Strømgaard and members of the Pless lab for helpful comments on the manuscript.

## Author contributions

K.K.K., I.G., and S.A.P. designed the research. K.K.K., I.G., F.G., R.W., H.H., M.H.P., H.C.C, M.W. performed the experiments. K.K.K., I.G., F.G., R.W., H.H., M.H.P., H.C.C., M.W. analyzed the data. R.T. and S.A.P. supervised the project. K.K.K., I.G., and S.A.P. wrote the manuscript with input from all authors.

## Competing interests

The authors declare no competing interests.
