## [Peer Review File · Nature Communications]

Reviewers' Comments:

Reviewer #1:

Remarks to the Author:

In this manuscript, Khoo et. al. utilize split inteins for the installation of post translational modifications on soluble and membrane proteins, further expanding the sphere of intein applications in in vivo contexts. The authors demonstrate this ability by simultaneously adding two PTMs on a linker portion of the Nav1.5 channel in oocytes and investigating the effects. The authors also use split inteins to semi-synthetically generate a modified GFP in HEK cells, and P2X2R channel, in both oocytes and HEK cells to probe the significance of the K71 position. Overall this work, while not overcoming all the obstacles of working with inteins, illustrates the value of orthogonal split inteins in protein engineering, especially in cellular contexts. I support the publication of this work in Nature Communications, after the following few points are addressed:

- 1) The authors use Cfa DnaE and Ssp DnaB as their intein pairs. While the authors suggest that these inteins are orthogonal, I didn't find any reference to support this. If there is a reference for that, the authors should mention and cite it and if there isn't, the authors should perform an experiment demonstrating there is no cross reactivity between the intein fragments.
- 2) The authors state they chose the Cfa and Ssp inteins because they can be split asymmetrically. However, they don't mention anything about the kinetics of splicing. CfaDnaE is relatively well characterized, but SspDnaB is not really and there is no reference to its kinetic characterization. It is not related to the SspDnaE, which is a slow splicing intein, but is it ultra-fast? How is it compared with Cfa? How does the kinetic of the different inteins affect the overall semi-synthesis? The authors don't really explain the reason for selecting these inteins for their strategy design. Since this manuscript is geared toward the general audience, it would benefit from a further discussing of this issue.
- 3) Throughout the paper the authors note the low yields of the spliced proteins they are trying to create. In the discussion they note a few reasons this could be the case, however, the manuscript would benefit from the authors investigating this issue a bit further. For instance, how can the authors be sure of the stoichiometries of the individual pieces? Perhaps the pieces have different stabilities? This could be addressed through epitope tagging on the expressed pieces and visualization via immunofluorescence and western blotting, but at minimum needs to be discussed further.
- 4) A brief explanation of the choice to use oocytes as the initial model, and a discussion on how this simplifies the use of inteins in cells (specifically the delivery of the synthetic moieties via injection rather than the more challenging techniques required in other cell types).
- 5) In the experiments relating to the P2X2 receptor, the work done in the oocytes vs. the HEK cells is not made clear. Again, for the benefit of the readership, the authors should further elaborate these selections. For example, Figure 6 and most of the text states that the work was done in oocytes, yet at the end of this section the authors note they were able to reconstitute P2X2 receptors in HEK cells. However, the main text describes an experiment using one split intein with no deliverable synthetic moiety. A more detailed explanation of the purpose of this experiment and what it offers that the work with the oocytes does not would be appropriate.

Reviewer #2:

Remarks to the Author:

The manuscript describes the use of a tandem protein trans-splicing (T-PTS) approach for protein modifications in live eukaryotic cells. The ability to modify the structural and electronic properties is a key requirement for protein structure-function studies. The approach of protein synthesis (or semi-synthesis) using ligation approaches is a very powerful approach for protein modification but is limited to in vitro studies. The discovery of split inteins was a major step forward in facilitating in vitro protein assembly and also for protein synthesis to be carried out in cells.

The manuscript represents a technical advance in that protein assembly is carried out from three

fragments using a tandem trans-splicing approach. The central fragment used is synthetic and so can be easily modified and these modifications are then incorporated into the protein under investigation. Another key advantage is that the process is being carried out in a live eukaryotic cell.

The authors establish the feasibility of the approach using the Nav1.5, GFP and the P2X2 proteins. Further the approach is used to introduce modest unnatural substitutions into the Nav1.5 and the P2X2 receptors to illustrate the utility of the approach.

There are a few points to consider:

- 1) Split inteins have sequence requirements at the splice sites and so the strategy described will result in insertion of a few amino acid residues at both the splice sites. These additional residues that are incorporated should be indicated in the scheme depicted in figure 1 and described clearly in the text. The approach is not traceless which can be a limitation for a number of applications.
- 2) The ligated Nav1.5 channel is mentioned to be functionally indistinguishable from the wild type. This is not correct as the authors state that the N1427C substitution shifts the inactivation profile. I think what the authors mean to say is that Nav1.5 channel obtained after the tandem protein trans-splicing approach is functionally similar to the corresponding recombinant control.
- 3) The blots in figure 2c do not look convincing for the generation of the full length channel. The blots in figure S2 are slightly better but the full length product appears to be migrating higher than the wild type control. This should be explained.
- 4) The yield of the GFP molecules generated using the T-PTS approach is likely to be very low. Immunoblots should be presented to provide an idea of the yield of the product obtained. While low yields are not an issue for ion channels, as functional studies can be carried out on a small populations of channels, the same does not hold true for cytoplasmic proteins. The low yields will make this approach not useful for cytoplasmic proteins. If the authors want to claim that the T-PTS approach can be used for cytosolic proteins then the approach has to be demonstrated on a protein other than GFP, for example on a signaling protein wherein change in the cellular status can be demonstrated.
- 5) In addition to the low yields, other limitations that have to be mentioned are that the approach cannot be used to modify transmembrane helices which contain key residues in ion channel proteins. The approach is only suitable for modifying linker regions in ion channels (and potentially with expression in *Xenopus oocytes*). The authors need to tone down the discussion of the scope of the approach.

In spite of these limitations, the approach has advantages in the number of modifications that can be introduced. Approaches such as using an amber codon and a four base pair codon allow the incorporation of two unnatural modifications while the T-PTS approach will allow multiple modifications in the region contained within the synthetic peptide. The approach described is likely to be useful for modifying linker regions in ion channels and therefore a useful addition to the toolkit for modifying ion channels.

The study is therefore likely to be of general interest for the readership of *Nature Communications*.

Reviewer comments:

Reviewer #1:

In this manuscript, Khoo et. al. utilize split inteins for the installation of post translational modifications on soluble and membrane proteins, further expanding the sphere of intein applications in in vivo contexts. The authors demonstrate this ability by simultaneously adding two PTMs on a linker portion of the Nav1.5 channel in oocytes and investigating the effects. The authors also use split inteins to semi-synthetically generate a modified GFP in HEK cells, and P2X2R channel, in both oocytes and HEK cells to probe the significance of the K71 position. Overall this work, while not overcoming all the obstacles of working with inteins, illustrates the value of orthogonal split inteins in protein engineering, especially in cellular contexts. I support the publication of this work in Nature Communications, after the following few points are addressed:

We would like to thank the reviewer for pointing out the value of our work for protein engineering in live cells.

1) The authors use Cfa DnaE and Ssp DnaB as their intein pairs. While the authors suggest that these inteins are orthogonal, I didn't find any reference to support this. If there is a reference for that, the authors should mention and cite it and if there isn't, the authors should perform an experiment demonstrating there is no cross reactivity between the intein fragments.

We thank the reviewer for raising this important point. We have now cited the reference supporting the orthogonality of these split inteins [Demonte, D., Li, N. & Park, S. Postsynthetic Domain Assembly with NpuDnaE and SspDnaB Split Inteins. *Applied biochemistry and biotechnology* **177**, 1137-1151, doi:10.1007/s12010-015-1802-0 (2015).] While we have used optimized versions of the split inteins in the referenced study, the negative results from our N+C controls also confirm that there is no substantial cross reactivity between these split inteins.

2) The authors state they chose the Cfa and Ssp inteins because they can be split asymmetrically. However, they don't mention anything about the kinetics of splicing. CfaDnaE is relatively well characterized, but SspDnaB is not really and there is no reference to its kinetic characterization. It is not related to the SspDnaE, which is a slow splicing intein, but is it ultra-fast? How is it compared with Cfa? How does the kinetic of the different inteins

affect the overall semi-synthesis? The authors don't really explain the reason for selecting these inteins for their strategy design. Since this manuscript is geared toward the general audience, it would benefit from a further discussing of this issue.

We agree with the reviewer and now discuss our choice of split inteins in more detail in the Discussion:

“Second, we note that numerous other split inteins³⁴ with different extein requirements could alternatively be used for this approach and could potentially, depending on the context, yield higher splicing efficiency. Here, we chose the split inteins CfaDnaE and SspDnaB^{M86}, as they have been well characterized with fast kinetics and engineered to have increased tolerance to non-native extein sequences^{10,11}. Importantly, SspDnaB^{M86} can be split asymmetrically with the Int^N segment only comprising 11 amino acids, making it an ideal split intein B in this approach (Fig 1).”

3) Throughout the paper the authors note the low yields of the spliced proteins they are trying to create. In the discussion they note a few reasons this could be the case, however, the manuscript would benefit from the authors investigating this issue a bit further. For instance, how can the authors be sure of the stoichiometries of the individual pieces? Perhaps the pieces have different stabilities? This could be addressed through epitope tagging on the expressed pieces and visualization via immunofluorescence and western blotting, but at minimum needs to be discussed further.

We agree with the reviewer that these issues would be worth investigating further. We have expanded our discussion to include these factors for consideration, see revised Discussion. However, given the large number of other possible contributing factors, which we explicitly outline in our discussion, we think that these issues would be better tackled in a separate study focusing on the optimization of this approach.

The discussion now includes the following section:

“Furthermore, it cannot be excluded that the recombinantly expressed protein fragments display different stabilities toward the proteasome or are differentially trafficked, resulting in unequal fragment ratios and thus potentially suboptimal conditions for splicing to occur. The length, proteolytic stability, and solubility of synthetic peptides, along with requirements for native-like flanking extein sequences, can also affect splicing efficiency and reaction rates³⁵. Additionally, the amount of synthetic peptide that can be delivered into a cell is typically

limited by the viability of the cell in response to delivery of the peptide and peptide concentration. Lastly, factors that contribute to optimal splicing conditions, such as pH or redox potential, which are controllable in vitro, are virtually impossible to manipulate in a live cell. Indeed, it is possible that the lower splicing efficiency we observed when using tPTS to modify the extracellular domain of the P2X2 receptor was due to unfavorable redox conditions in the endoplasmic reticulum and/or the low abundance of synthetic peptide in this subcellular compartment (or others that the splicing could take place in)."

4) A brief explanation of the choice to use oocytes as the initial model, and a discussion on how this simplifies the use of inteins in cells (specifically the delivery of the synthetic moieties via injection rather than the more challenging techniques required in other cell types).

We have now included a explanation for using *Xenopus laevis* oocytes as an initial model system and expanded our discussion to include considerations for applications in other cell types.

In the Results section:

"These constructs were transcribed into mRNA and injected into Xenopus laevis oocytes for recombinant expression. This approach is well-established for assessing ion channel function using electrophysiology and, conveniently, allows for direct delivery of mRNA and/or peptides into the cytosol using microinjection²⁶."

In the Discussion:

"Finally, the means of introducing the synthetic peptide X needs to be optimized depending on the cell type in question. While synthetic peptides can be injected directly into Xenopus laevis oocytes, our approach requires potentially more challenging delivery techniques, such as cell squeezing, electroporation or the use of cell-penetrating peptides when implemented in mammalian cells."

5) In the experiments relating to the P2X2 receptor, the work done in the oocytes vs. the HEK cells is not made clear. Again, for the benefit of the readership, the authors should further elaborate these selections. For example, Figure 6 and most of the text states that the work was done in oocytes, yet at the end of this section the authors note they were able to reconstitute P2X2 receptors in HEK cells. However, the main text describes an experiment

using one split intein with no deliverable synthetic moiety. A more detailed explanation of the purpose of this experiment and what it offers that the work with the oocytes does not would be appropriate.

Thank you for pointing this out. The main purpose of reconstituting P2X2 receptors in HEK cells was to demonstrate that splicing within extracellular domains could take place in both *Xenopus laevis* oocytes and mammalian cells. We have now mentioned this more explicitly in the text.

Reviewer #2 (Remarks to the Author):

The manuscript describes the use of a tandem protein trans-splicing (T-PTS) approach for protein modifications in live eukaryotic cells. The ability to modify the structural and electronic properties is a key requirement for protein structure-function studies. The approach of protein synthesis (or semi-synthesis) using ligation approaches is a very powerful approach for protein modification but is limited to in vitro studies. The discovery of split inteins was a major step forward in facilitating in vitro protein assembly and also for protein synthesis to be carried out in cells.

The manuscript represents a technical advance in that protein assembly is carried out from three fragments using a tandem trans-splicing approach. The central fragment used is synthetic and so can be easily modified and these modifications are then incorporated into the protein under investigation. Another key advantage is that the process is being carried out in a live eukaryotic cell.

The authors establish the feasibility of the approach using the Nav1.5, GFP and the P2X2 proteins. Further the approach is used to introduce modest unnatural substitutions into the Nav1.5 and the P2X2 receptors to illustrate the utility of the approach.

We would like to thank the reviewer for pointing out utility and key advantages of our work.

There are a few points to consider:

- 1) Split inteins have sequence requirements at the splice sites and so the strategy described will result in insertion of a few amino acid residues at both the splice sites. These additional residues that are incorporated should be indicated in the scheme depicted in figure 1 and described clearly in the text. The approach is not traceless which can be a limitation for a number of applications.

We thank the reviewer for raising this important point. We have now made the suggested changes: Figure 1 now indicates the critical sequence requirements for successful splicing and we mention how this could be a limitation in other applications in the Discussion.

“In cases where the native sequence does not contain residues required for splicing, mutations may need to be introduced at the intended splice sites to fulfil the extein requirements for successful splicing (i.e. the need for a Cys or Ser at the +1 extein position of the extein, see Fig 1).”

2) The ligated Nav1.5 channel is mentioned to be functionally indistinguishable from the wild type. This is not correct as the authors state that the N1427C substitution shifts the inactivation profile. I think what the authors mean to say is that Nav1.5 channel obtained after the tandem protein trans-splicing approach is functionally similar to the corresponding recombinant control.

We believe the reviewer is referring to the following original sentence:

“Remarkably, co-injection of mRNA corresponding to $N+C+X_{Nav1.5}^{REC}$ resulted in full-length channels that showed robust current levels and were functionally indistinguishable from the full-length, recombinant N1472C mutant (Fig 2d-e).”

We can confirm that we did compare the ligated Nav1.5 channel ($N+C+X_{Nav1.5}^{REC}$) to the N1472C mutant, as written in the original text. However, and in order to avoid ambiguity, we have reworded the sentence to:

“Remarkably, co-injection of mRNA corresponding to $N+C+X_{Nav1.5}^{REC}$ (i.e. containing the N1472C mutation) resulted in full-length channels that showed robust current levels and were functionally indistinguishable from the full-length, recombinantly expressed channel construct also bearing the N1472C mutation (Fig 2d-e).”

3) The blots in figure 2c do not look convincing for the generation of the full length channel. The blots in figure S2 are slightly better but the full length product appears to be migrating higher than the wild type control. This should be explained.

We agree that the bands corresponding to the full length channel in Figure 2c are very faint, indicative of the low amounts of product formed, which we mention in the text. An overexposed version of the blot (see below; also available in the Source Data file) confirms

that the bands representing fully spliced products are present only when all three splicing competent constructs are used, but not in any of the control conditions, confirming that fully spliced product (with a MW equal to the full-length WT protein) is indeed formed.

Blot as shown in manuscript:

Same blot, but imaged with increased exposure time:

We presume 'the blots in figure S2' that the reviewer refers to are the blots shown in Figure S4. We note that in Figure S4c, the band corresponding to the spliced full length product might appear slightly higher than the WT control. However, we did not consistently see this pattern (see Figure S4h) and the observation is likely due to the slightly tilted blot (the uncropped blot will be made available in a Source Data file on Zenodo.com).

4) The yield of the GFP molecules generated using the T-PTS approach is likely to be very low. Immunoblots should be presented to provide an idea of the yield of the product obtained. While low yields are not an issue for ion channels, as functional studies can be carried out on a small populations of channels, the same does not hold true for cytoplasmic proteins. The low yields will make this approach not useful for cytoplasmic proteins. If the authors want to claim that the T-PTS approach can be used for cytosolic proteins then the approach has to be demonstrated on a protein other than GFP, for example on a signaling protein wherein change in the cellular status can be demonstrated.

We appreciate this point raised by the reviewer and agree that with the low efficiency observed, the approach is mostly limited to applications or proteins that can be detected with high-resolution techniques. We now mention these limitations more explicitly and have toned down the discussion regarding the scope of the approach. Concerning the splicing efficiency

of the reconstitution of eGFP, we performed additional FACS experiments to quantify the efficiency of GFP reconstitution, which is now shown in the new Fig. S6.

5) In addition to the low yields, other limitations that have to be mentioned are that the approach cannot be used to modify transmembrane helices which contain key residues in ion channel proteins. The approach is only suitable for modifying linker regions in ion channels (and potentially with expression in *Xenopus* oocytes). The authors need to tone down the discussion of the scope of the approach.

We thank the reviewer for pointing this out. We have now extensively expanded the Discussion section to elaborate on the limitations and considerations for the use of the approach in other contexts. We now also mention explicitly that changes in transmembrane helices will not be possible with the present approach.

In spite of these limitations, the approach has advantages in the number of modifications that can be introduced. Approaches such as using an amber codon and a four base pair codon allow the incorporation of two unnatural modifications while the T-PTS approach will allow multiple modifications in the region contained within the synthetic peptide. The approach described is likely to be useful for modifying linker regions in ion channels and therefore a useful addition to the toolkit for modifying ion channels. The study is therefore likely to be of general interest for the readership of Nature Communications.

We would like to thank the reviewer for the positive remarks on the suitability of our work for publication in Nature Communications.

Reviewers' Comments:

Reviewer #1:

Remarks to the Author:

The authors have answered all of my questions and have provided additional explanations/clarifications where required. I am happy to accept this revised version.

Reviewer #2:

None